# Marital Satisfaction and Perceived Family Support in Families of Children with Autistic Spectrum Disorder: Dyadic Analysis

**DOI:** 10.3390/healthcare10071227

**Published:** 2022-06-30

**Authors:** Bijing He, Tinakon Wongpakaran, Nahathai Wongpakaran, Danny Wedding

**Affiliations:** 1Graduate School, Chiang Mai University, Chiang Mai 50200, Thailand; bijing_he@cmu.ac.th (B.H.); nahathai.wongpakaran@cmu.ac.th (N.W.); danny.wedding@gmail.com (D.W.); 2Department of Psychiatry, Faculty of Medicine, Chiang Mai University, Chiang Mai 50200, Thailand; 3Department of Clinical and Humanistic Psychology, Saybrook University, Pasadena, CA 91103, USA; 4Department of Psychology, University of Missouri-Saint Louis, St. Louis, MO 63121, USA

**Keywords:** family support, autism, couple satisfaction, parents, actor partner interdependence model

## Abstract

Background: Raising children with autism spectrum disorder (ASD) causes tremendous stress for parents that may lead to marital conflict and relationship dissatisfaction. Many factors are associated with parent relationships including severity of autistic behaviors and social support. This study aimed to investigate whether severity of autistic behaviors, perceived family support, and complementarity of interpersonal styles between husbands and wives predicted couple satisfaction among the parents of children with ASD. Method: Seven hundred ninety-seven parent dyads of children aged 7–14 years old with ASD participated in the study. Measurements used included couple satisfaction index, perceived family support using the Multidimensional Scale of Perceived Social Support, inter-personal style using the inventory of interpersonal problems, the ABC autism checklists as well as sociodemographic and related factors. The Actor Partner Interdependence Model estimated by multilevel modeling was used for analysis. Results: Perceived family support was relevant in married couples regarding their marital relationship, but the effects on husbands and wives differed. Husbands’ relationship satisfaction was predicted by how they perceived being supported by family. The severity of autistic behaviors predicted relationship satisfaction but only actor effect. Negative prediction of interpersonal complementarity on couple satisfaction was observed. In addition, time spent on raising children had a negative impact on the quality of the relationship. Partner effect of time spent was observed among women. Conclusion: Dyadic analysis using an actor–partner independence model confirmed perception of family support predicts relationship satisfaction among parents of children with ASD in addition to the severity of autistic behaviors and time spent caring for children. Complementarity of individual interpersonal style had no effect on couple satisfaction. This research suggests implications for interventions regarding building skills that elicit support from family members.

## 1. Introduction

Autistic spectrum disorder (ASD) is a group of developmental disorders involving social communication disorder syndrome with repetitive and rigid behaviors that develop in early childhood [1]. It remains a lifelong disability, so special care may be needed. Having a child with ASD in the family causes tremendous stress for parents [2] as caring for children with ASD may affect all aspects of family life, including housework, finance, parents’ emotional and mental health, family functioning, and the marital relationship [3,4]. Parents of children with ASD have lower satisfaction with relationships than parents of children without disabilities [5]. Due to the challenges in such families, many couples find it difficult to overcome this hurdle. Research has shown that approximately 24% of parents in families with ASD ended up in divorce [6]. One possible explanation is that child severity of autistic behaviors was related to higher levels of parenting stress and coparenting conflicts, which, in turn, resulted in increased marital conflicts and decreased marital satisfaction among parents of children with ASD [7].

### 1.1. Importance of the Marital Relationship

The relationship between parents of children with ASD has been the focus of numerous research studies. One study demonstrated that the relationship between parents declined after the birth of a child with ASD [8]. It has become evident that parents of children with ASD have a low level of relationship satisfaction, which was unrelated to socioeconomic status [9,10]. Lower marital satisfaction influences parenting burden and the quality of the parent–child relationship [8,11], quality of child caregiving and also child externalizing and internalizing symptoms [12,13,14,15], while a good marital relationship is an important resource for dealing with the difficulty in the challenges of raising a child with ASD.

### 1.2. Impact of Perceived Family Support

Perceived family support has played an important and crucial role as a resource in the lives of people with mental illness [16,17]. Parents of children with ASD often become fearful and feel out of control, requiring support [18,19]. One study demonstrated that perceived family support could reduce the effect of child’s severity of autistic behaviors on parental mental health [20]. Family support was related to parental competency, and greater partner family functioning [21], through parental resilience and self-efficacy, resulting in reducing the emotional/behavioral problems among children with ASD [22]. Support resulted in decreasing children’s behavioral problems, strengthened the parental relationship [23], and promoted life satisfaction [18].

However, one study demonstrated a positive effect of spousal support on individuals and relationship satisfaction [24] in a collectivistic culture like China, where family members include parents, children, grandparents and other relatives [25]. Particularly, grandparents play a huge role in childcaring in this culture [26]. The authors believe that viewing total support in a family is more useful rather than focusing only on spousal support.

### 1.3. Other Associated Factors: Severity of Autistic Behaviors and Interpersonal Complementarity between Couples

Parent couple relationship satisfaction has been found to be negatively associated with the severity of challenging behaviors a child [9]. Greater challenging behaviors results in more stress among parents, which, in turn, resulted in increased marital conflicts and decreased marital satisfaction among parents of children with ASD [7]. In addition to the severity of autistic behaviors that have an impact on the couple’s relationship, interpersonal style and compatibility between parents may determine how well the couple could overcome conflicts when encountering stressful situations. Few studies have investigated the interaction between personality traits or interpersonal styles. One study explored personality synchrony and similarity among couples, and found that similarity in neuroticism, extraversion, and openness predicted perceived spousal support. However, the effects of similarity were relatively small compared with actor and partner effects of these personality traits [27]. In contrast, interpersonal complementarity has been explored revealing that complementarity was more similar in terms of warmth but more dissimilar in terms of dominance in predicting high levels of relationship harmony [28]. In the present study, the authors were interested in examining how the relationship satisfaction levels were related to interpersonal complementarity among parents of children with ASD. To date, few studies have investigated the relevance of perceived family support of a partner for women’s and men’s couple satisfaction. One study demonstrated that social and familial support is associated with positive marital relationships assessed by individual partners [29]. It became evident, however, that satisfaction related to family support has a dyadic nature because the way partners feel affects how individuals experienced stress in their families. Thus, dyadic analysis involves using an appropriate approach. This article aims to investigate the predicting effects of perceived family support, severity of autistic behaviors, interpersonal complementarity and other significant sociodemographic variables on couple satisfaction. The authors hypothesized that perceived family support and interpersonal complementarity should exhibit a positive association, but severity of autistic behaviors should show a negative association with satisfaction.

Moreover, the actor–partner interdependence model (APIM) [30], a model allowing the study of the effects of transactions between two individuals (husbands and wives) within the same dyad (couple or family unit) should be able to account for the interdependence of observations [31]. The hypotheses that can be tested for each effect yielded by the APIM, i.e., actor effect, partner effect, and interaction effect, are presented below (Figure 1). Using APIM, we examine the impact of spouses’ perceptions of family support on their mutual satisfaction with the relationship, while controlling the perception of being supported as experienced by partners.

## 2. Materials and Methods

The research employed a cross-sectional survey, conducted online between December 2021 and January 2022 due to COVID-19 and the lockdown policy in mainland China. This research was approved by the Research Ethics Committee, Faculty of Medicine, Chiang Mai University (IRB number: 20211220). Informed consent was obtained from all participants in the study.

### 2.1. Participants

Participants who were the parents of children with ASD were recruited from the general population all over mainland China. The inclusion criteria included (1) participants taking care of child(ren) with ASD and residing together as couples at the time of conducting the research, (2) each couple having at least one child with ASD in the family, (3) the diagnosis of ASD for the child(ren) had to be made by the doctor(s) in a hospital, and each couple was asked to attach the document regarding medical records issued by the hospital for the researcher, (4) participants were fluent in Chinese and able to access the Internet online survey, and finally (5) each participant was able to complete the questionnaires on one’s own and independently.

### 2.2. Procedure

All participants were provided with a detailed description of the study and the intended use of the results. Each couple was provided with two surveys and each spouse filled out their own independently. Questionnaire data were kept confidential to protect the identity of participants. They were asked to create theirs’ and their partners’ first-letter initials of first name and last name, including date of birth for data matching. All participants were volunteers and did not receive any monetary compensation.

A related study reported that 3% of the subjects in the population have the factor of interest, so the study would require a sample size of at least 280 pairs to estimate the expected proportion with 2% absolute precision and 95% confidence. In other words, a random sample of 280 pairs from a population that could determine that 3% of pairs having the factor of interest would constitute the 95% confidence level between 1 and 5% of pairs in the population had the factor of interest [32].

A total of 1030 volunteers took part in the questionnaire. Among all, 24 couples were excluded because their children with ASD were less than 7 years old or more than 14 years old, and 212 other participants were excluded because they were not able to match with the other couple. Consequently, we checked the first letter initials of each participant’s first and last name, plus the date of birth (including year, month and day), to match his or her partner. The final sample included 397 couples (total *n* = 794).

### 2.3. Measurements

#### 2.3.1. Demographic Data of Parents’ and Children’s Information

Participants provided basic demographic and related data. These included sex, age, education level, income, socioeconomic status, number of children, residence area (urban or rural), marital status, duration of marriage, and type of marriage (arranged or self- deliberated) and time spent for daily caregiving of children.

#### 2.3.2. Multidimensional Scale for Perceived Social Support

The Multidimensional Scale for Perceived Social Support (MSPSS) is one of the most widely used tools for evaluating how an individual feels about being socially supported [33]. The scale applies a 7-point Likert scale (1 = very strongly disagree; 7 = very strongly agree). The total scale ranges from 12 to 84, and high scores equate to a high perception of social support. A score of 12 to 36 indicates low support, 37 to 60 indicates moderate support, and 61 to 84 indicates high support [33]. The Chinese version of the MSPSS was demonstrated to be reliable and valid [34]. It has three subscales including family support, friend support, and other major support. The Cronbach coefficients for the family, friends, and other important support subscales were 0.88, 0.89, and 0.87, respectively [34]. In the current study, only family support was used.

#### 2.3.3. Autism Behavior Checklist (ABC)

The ABC is a 57-item behavioral rating scale that evaluates autistic behaviors and symptoms among children 3 years of age and older. The tool consisted of 57 questions and is divided into five categories: (1) sensation, (2) correlation, (3) use of body and objects, (4) language, and (5) society and self-help. The weighted score for each term ranged from 1 to 4. The total score ranges from 0 to 158 and high scores on the total scale indicate greater levels of autism symptoms. The score of 68 is used as the cut-off value for considering positive for autistic spectrum disorder. ABC sessions are designed to be completed independently by parents or teachers who are familiar with the child for a minimum of 3 to 6 weeks and should take 10 to 20 min to complete [35]. The Chinese version of the ABC was reliable and valid [36]. The Cronbach coefficient in this sample was 0.95.

#### 2.3.4. Couple Satisfaction Index (CSI)

The CSI is a 16-item scale measuring how satisfied a person is with a relationship [37]. The CSI-16 involves a 6-Likert type response, from 0 (not at all true) to 5 (completely true). Scores can range from 0 to 80. The sample questions included, “Our relationship is strong”and “My relationship with my partner makes me happy”. Higher scores indicate higher levels of relationship satisfaction. A score below 51.5 indicates significant dissatisfaction [38]. The couple satisfaction scale has been translated into Chinese and revised in China. Cronbach’s alpha of the Chinese version of the CSI was 0.93.

#### 2.3.5. Inventory for Interpersonal Problems (IIP)

The IIP is a self-report tool, aiming at assessing problems or style in the field of interpersonal communication, which are reflected in the difficulties in implementing specific behaviors [39]. This tool is based on the common interpersonal behavior theory which has a long tradition in the field of personality and social psychology [40]. It consists of 32 questions including eight different interpersonal styles or problems: domineering/control (DO), retaliation/self-centeredness (VI), cold/alienation (CO), social inhibition (SI), nonassertion (NO), overly-accommodating (OA), self-sacrifice (SS), and intrusion/need (IN) [41,42]. Each scale has four items. The total score ranges from 0 to 128. A higher score indicates a greater difficulty in interpersonal communication. The cut-off of the t-score ranges from 40 to 60, below 40 is low, above 60 is high. The Chinese version of IIP-32 showed a Cronbach’s Alpha coefficient between 0.68 and 0.81 [43]. In this study, Cronbach’s Alpha coefficient of the eight subscales ranged from 0.70 to 0.84.

#### 2.3.6. Interpersonal Complementarity

We employed interpersonal complementarity as the interpersonal relationship matching for this research [40]. The complementarity would not be considered favorable or unfavorable interactional outcome of the dyad, but it meant that one style would evoke a particular interpersonal response of the other. To determine the levels of complementarity or noncomplementarity, models proposed by Carson, Byrne, and Wiggins were used.

Carson’s model of interpersonal complementarity explains that individuals, similar to each other concerning warmth, but opposite concerning dominance, are the most compatible [44], Byrne’s similarity model clarifies that individuals with similar personalities are the most compatible [45], Wiggins’s model of interpersonal complementarity describes that individuals whose personalities occur in a manner, predicted by social exchange theory, are the most compatible [46,47,48]. For example, domineering husband and submissive wife were considered complementary according to Carson’s model [44], whereas domineering husband and warm-agreeable wife were considered matched according to Wiggins’s model [49]. However, domineering husband and aloof-introverted wife were considered noncomplementary. Each model was related to different stages of a relationship, e.g., romantic ideals, romantic obtainment, and relationship experiences [28].

In this research, the authors used all proposed models to determine whether the couple conveyed interpersonal complementarity or not.

### 2.4. Statistical Analysis

Descriptive statistics and correlation analysis were reported for husbands and wives, respectively, using IBM SPSS, 22.0. Potential covariates can be discovered by analyzing possible group differences according to demographic characteristics. Missing data were handled by an expectation–maximization method. The main assumptions in this study were tested using the actor–partner interdependence model [50], to assess maternal and paternal perceptions of family support in relation to couple satisfaction. Perceived family support was the independent variable, and couple satisfaction was the dependent variable, calculated separately for husbands and wives. The APIM enabled us to test both how parents’ predictive variables affected their own (actor effect) and their partner’s (partner effect) outcomes [51]. The analyses employed generalized least squares analysis with correlated errors and restricted maximum likelihood estimation. The tests of coefficients are Z tests and the tests of correlations are based on *t*-tests of correlation coefficients. Effect sizes for actor and partner effects are partial correlations and ‘d’ when the predictor is dichotomous. Betas are given twice, one using the overall standard deviation across all persons (o) and a second using the standard deviation for husbands and wives separately(s). If betas are to be compared across members, the beta (o) value should be examined. All predictors were grand-mean-centered before the analysis. The partial correlations between predictor and outcome variables, controlling for all other predictors, were calculated as effect sizes. Values above r = 0.10 indicated a small effect size, between r = 0.30 and r = 0.50 a medium effect size and above r = 0.50, a large effect size [52].

In addition, in APIM analysis, the different patterns of interdependence were tested using k parameters (a ratio of the partner effect on the actor effect) [53]. The four patterns are: (1) actor-only pattern, when a k parameter with a value is near 0, and a ≠ 0, *p* = 0, (2) partner-only pattern, when a k parameter with a value is near 0, and a = 0, *p* ≠ 0, (3) couple-oriented pattern, when a k parameter is near 1, and a = p, and finally (4) contrast pattern, when a k parameter is near −1, and (a + p = 0) [54]. The APIM using multilevel modeling, written by David A. Kenny, was used for dyadic analysis in this study. The computer software was a web-based interactive tool created by Kenny [55]. For all analyses, alpha was set at 0.050.

## 3. Results

### 3.1. Sociodemographic Results

Among 397 couples (794 participants in total) with parents ranging in age from 23 to 45, no significant difference was observed in mean age between fathers (M = 36.33, SD = 3.36) and mothers (M = 35.36, SD = 3.07). The majority had university degrees, were employed (94.1%), earned 5000 to 10,000 CNY for family monthly income, and resided in urban areas (68.3%). Most families had one child (71.5%), aged 7 to 14 years, with 63.2% boys. Most families spent up to 30,000 CNY yearly for the cost of raising a child and spent time in raising a child up to 6 h daily. In terms of marriage, almost all reported being in the first marriage. Over one half fully made their decision to choose their own partner (57.9%) and had been married couples for more than 10 years (Fathers 88.4% and mothers 87.9%). In terms of interpersonal complementary, most were complementary (91.2%). No statistical differences regarding the scores of ABC perceived family support and couple satisfaction were observed. Details are shown in Table 1.

### 3.2. Correlation Results

Regarding correlation between couple satisfaction and other variables between husbands and wives, age of married couples, monthly income, time spent for raising a child, and type of marriage were significantly related (*p* < 0.05). ABC and perceived family support scores were significantly related to couple satisfaction (*p* < 0.01).

### 3.3. Summary of APIM Results with Distinguishable Dyads

Before testing the APIMs, preliminary analyses were carried out to determine whether the inclusion of covariates was required. Table 2 shows some significant variables that were associated with the outcome (couple satisfaction) of both husbands and wives that should be included in the APIM model.

The focus of this study was the investigation of the effect of income, time spent, severity of autistic behaviors, and family support on couple satisfaction. The dyad members were distinguishable by their role in marriage.

Three covariates for the couples were included in the analysis, i.e., residence area, type of marriage and interpersonal complementarity, whereas monthly income, time spent for caregiving, severity of autistic behaviors, and perceived family support was controlled for individual. There were four outliers in the dataset which were removed from the analysis.

The proportion of the total variance explained by the actor and partner variables after controlling for covariates for husbands was 0.25 and for wives was 0.25 (*p* < 0.001). The intercept for husbands was 82.85 and significantly differed from zero (*p* < 0.001), and the intercept for wives was 60.79 and was statistically significant (*p* < 0.001). The difference between the two, which is a test of the main effect of the role in marriage, was not statistically significant (*p* = 0.059). The overall intercept was 71.82 and significantly differed from zero (*p* < 0.001) (Table 3).

#### 3.3.1. Severity of Autistic Behaviors

The actor effect for husbands equaled −0.64 and was statistically significant (*p* < 0.001), and the standardized effect equaled −0.31 (r = −0.31 with a medium effect size). The actor effect for wives equaled −0.33 (*p* < 0.001), and the standardized effect equaled −0.16 (r = −0.19 with a small effect size). The test where the two actor effects differed was significant, Z = 2.24 (*p* = 0.026). The partner effect from wives to husbands equaled 0.07 (*p* = 0.404); likewise, the partner effect for husbands to wives equaled 0.11 (*p* = 0.235) (Table 3).

The combined actor effect across both husbands and wives equaled −0.49 (*p* < 0.001), and the standardized effect equaled −0.23 (r = −0.25). The combined partner effect across both husbands and wives equaled 0.09 (*p* = 0.149). The overall k, the ratio of the partner effect to the actor effect, equaled −0.19 (95% CI, −0.50 to 0.07), suggesting that the actor-only model (k = 0) was plausible (Table 4).

#### 3.3.2. Family Support

The actor effect for husbands equaled 1.14 (*p* < 0.001) and the standardized effect equaled 0.38 (r = 0.41 and a medium effect size). The actor effect for wives equaled 1.47 (*p* < 0.001), and the standardized effect equaled 0.50 (r = 0.48 and a medium effect size). Two actor effects were not significantly different, Z = 1.76 (*p* = 0.079). The partner effect from wives to husbands equaled 0.34 (*p* = 0.015), and the standardized effect equaled 0.11 (r = 0.13 and a small effect size). The partner effect for husbands to wives equaled −0.01 (*p* = 0.948). The two partner effects were not significantly different, Z = −1.82 (*p* = 0.070) (Table 3 and Figure 2).

The combined actor effect across both husbands and wives equaled 1.31 (*p* < 0.001) and the standardized effect equaled 0.40 (r = 0.45 and a medium effect size). The combined partner effect across both husbands and wives equaled 0.16 (*p* = 0.082). The overall k, the ratio of the partner effect to the actor effect, equaled 0.13 (95% CI, −0.02, 0.27), suggesting that the actor-only model (k = 0) was plausible (Table 4).

#### 3.3.3. Time Spent for Caring Children

The actor effect for husbands equaled −2.22 (*p* < 0.001), for wives −1.85 (*p* < 0.001), whereas the partner effect for wives to husbands was significant (*p* = 0.046), but not for husbands to wives. The couple model (k = 1) was plausible (Table 3).

#### 3.3.4. Income

The actor effect for husbands equaled −2.29 (*p* = 0.003) and for wives −2.75 (*p* < 0.001). The partner effect both from wives to husbands and husbands to wives was not statistically significant (*p* = 0.660). The partner effects were not statistically significant for both husbands and wives (Table 3). The combined actor effect across both husbands and wives equaled −2.52 (*p* < 0.001). The relative sizes of actor and partner effects suggested that the actor-only model (k = 0) was plausible (Table 4).

#### 3.3.5. Covariates

The proportion of variance uniquely explained by the covariates for husbands was 0.01 and for wives was 0.02. The test of the null hypothesis that the variance uniquely explained by the covariates was zero yielded a chi square statistic with six degrees of freedom, equaling 21.14 (*p* = 0.002). Because this test was statistically significant, we concluded that that variance was statistically greater than zero. The covariate residence area was a between-dyads variable. Its overall effect across both husbands and wives equals −2.17 (*p* = 0.004). The effects of residence area for wives were −2.17 (*p* = 0.004, for husbands was −1.76 (*p* = 0.094). There was no significant difference in satisfaction for husbands and wives (Z = −0.55, *p* = 0.580). The effect of type of marriage on couple satisfaction for husbands was significant (Z = 2.05, *p* = 0.040), but not for wives (Z = 1.49, *p* = 0.130). The effect of interpersonal complementarity (Similarity model) for husbands equaled −1.35 (*p* = 0.260) and for wives 2.33 (*p* = 0.055). Two effects were significantly different, Z = 2.15 (*p* = 0.032). It can be concluded that there is a difference of the effects of similarity of interpersonal complementarity on couple satisfaction for husbands and wives. Notably, no significant effect was observed for Carson’s and Wiggins’ model.

## 4. Discussion

The purpose of this study was to examine how perceived family support and other relevant factors were associated with couple satisfaction among the parents of children with ASD at school age in China. The results of the current study have improved our understanding of how perceived family support and other related factors affect the satisfaction of both parents and are of great significance to the intervention plan for the parents of children with ASD. To the best of our knowledge, the current study is the first to examine the effects of perceived social support from family on relationship satisfaction among parents raising a child with ASD. Consistent with some related research, feelings of spousal support positively predicted how married couples felt about their marital relationship [24]. However, the present study focused on the whole family support rather than from spouses as in the aforementioned study.

Notably, the present study did not demonstrate how individual’s coping of seeking emotional support (for moral support, sympathy or understanding) influenced relationship satisfaction [56] but instead showed how married couples perceived the support from family members, which may play a more important role than the receiving of actual instrumental support. Our findings showed that such perceptions of receiving family support differ between husbands and wives. The fact that husbands’ satisfaction was predicted by both the husbands’ and the wives’ feelings of being supported suggested an unequal role in care provided in the family. It could be that the couple-oriented effect has been demonstrated among wives because wives (or mothers) are usually the primary care provider, particularly in Chinese culture [57]. As providing care creates stress, dyadic coping mediated the association between parenting stress and couple relationship satisfaction [10]. This study illustrated how receiving support from one’s family could be associated with satisfaction in a marital relationship. This finding was supported by other related studies [18,23].

Time spent raising children had a negative impact on the quality of relationship as it straightforwardly reduced the time parents can devote to their personal relationship. Interestingly, the time use is not only an actor effect, but it exhibited a partner effect on husbands. In other words, the longer time spent by wives had no effect on husbands’ relationship; in fact, the time spent for children by wives made husbands dissatisfied as well. This can be explained by the fact that husbands were unhappy because wives spent so much time with their children that they may neglect their husbands, and boys with ASD outnumber girls by nearly three to one [58].

In line with other related studies, severity of autistic behaviors had an effect on couple satisfaction [5]. However, the study demonstrated that only the actor affected how the parental partner perceived severity of autistic behaviors having no effect on the actor relationship satisfaction.

This study also adds new knowledge about the effect of interpersonal complementarity on couple satisfaction in that it exhibited a nonsignificant association. Complementarity indicates that the interpersonal style of each of them stimulates each other. One person’s unique interpersonal qualities tend to invite or pull a particular response from the other person [40]. Based on the model of complementarity used in this study, 12 of 18 complementary models (67%) of each couple expressed a similar style. The results suggested that the complementarity aspect may invite negative reactions, resulting in dissatisfaction in the relationship among parents of children with ASD. The fact that similarity style draws negative rather than positive reactions from the other partner could be because most of the similar styles fell within negative affiliation. We found that 65.8% of husbands and 70.8% of wives possessed the styles between domineering and non-assertive (Table 1). However, the effect was nonsignificant denoting that this interpersonal complementarity might be overshadowed by other factors

Regarding income, different associations of level of income and couple satisfaction were found between actor and partner of husbands and wives. Finally, the APIM model demonstrated a negative relationship on combined actor effect, suggesting that a higher level of income indicated a lower satisfaction in the marital relationship. These interesting results cannot be simply explained. It might be involved with other intervening factors that need to be further investigated.

In contrast to other related research, no effect of family income was observed. While many researchers have found that family income has a significant impact on marital satisfaction [5,59], some did not [9]. Notably, neither family or individual income predicted couple satisfaction, suggesting more weight was given to psychological or mental health issues.

### 4.1. Implications for Research

Being supported by one’s partner and other family members is important for parents who have children with ASD [60]. It is important to help married couples enjoy a beneficial and satisfying relationship, and to use these relationships as a resource in dealing with the stress of parenting. However, either providing or receiving family support might not be an easy task for everyone. Such skills can be taught and learnt, and strategic intervention on building effective family support can be implemented. Nonetheless, parents of children with ASD might have their own psychiatric vulnerability [20], which needs parental education and family support from external sources at the beginning before learning to cultivate it on their own. Moreover, adjusting time spent for children to act more appropriately, especially those with severe autistic behaviors, would help improve the relationship between parents.

### 4.2. Strengths and Limitations of This Study

The present investigation constituted one of the first studies to examine how perceived family support, severity of autistic behaviors, and interpersonal complementarity issues are linked to couple satisfaction using APIM to account for the importance of interdependencies.

Although this study presents some provocative findings, some limitations warrant discussion. First, this study was conducted among married and currently living parents, so extending these findings to other family structures might be unwise. Second, the study took place with Chinese married couples in mainland China, which might present different cultural and social values from Chinese in different parts of the world as well as people from western cultures. Finally, the cross-sectional nature of the research limits inferences about causality. Longitudinal research is needed to confirm the authors’ hypotheses.

## 5. Conclusions

This study used a dyadic model to examine how perception of family support was associated with relationship satisfaction among parents of children with ASD. An individual’s feelings of being supported, severity of behaviors, and time spent for caring of children were associated with degree of relationship satisfaction. Complementarity of individual interpersonal style had no effect on couple satisfaction. This research suggests the need for new interventions, for example, teaching mothers and fathers to reach out for emotional support, and to build up skills in obtaining support from family members, which may result in increasing the satisfaction of both members of the relationship.

## Figures and Tables

**Figure 1 healthcare-10-01227-f001:**
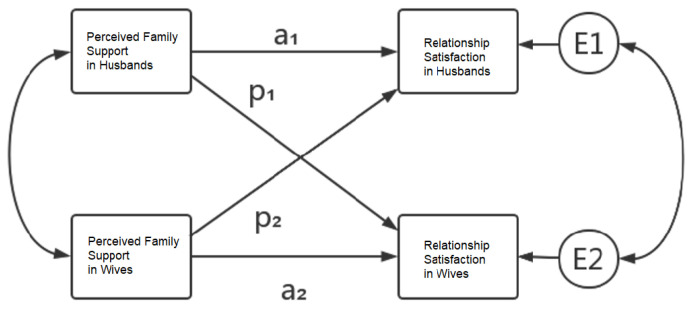
A hypothetical actor–partner interdependence model (APIM) with distinguishable members. The boxes on the left indicate the independent variables for husbands and wives. The boxes on the right indicate the couple satisfaction for each. E1 and E2 represent the residual error of couple satisfaction for husbands and wives, respectively. Single-headed arrows indicate predictive paths in which the husband’s participant (a_1_) and partner (p_1_) effects differ from the wife’s participant (a_2_) and partner (p_2_) effects. Paths labeled with ‘a’ indicate actor effects, and paths labeled with ‘p’ indicate partner effects. Double-headed arrows indicate correlated variables.

**Figure 2 healthcare-10-01227-f002:**
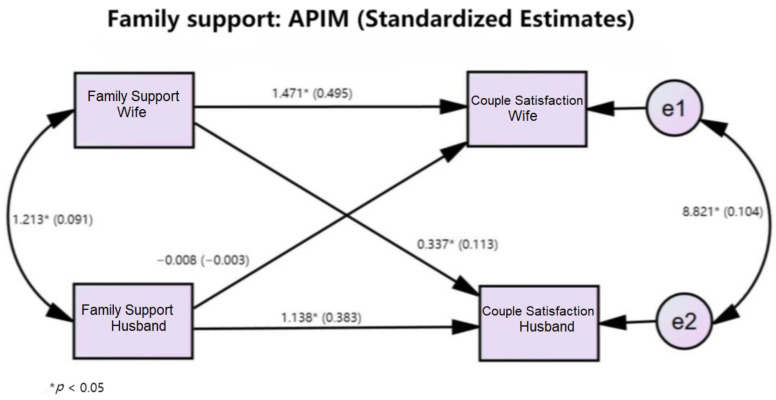
Actor–partner interdependence model for perceived family support.

**Table 1 healthcare-10-01227-t001:** Sociodemographic characteristics of the participants.

Variables	Husbands (*n* = 397)	Wives (*n* = 397)	Family (*n* = 397)	Test Difference
Age (Years), Mean (SD)	36.33 (3.36)	35.36 (3.07)		t = 4.29, *p* < 0.001
Age of first child (Years), Mean (SD)			9.93 (1.45)	
Sex of the first child with ASD *n* (%)				
Male			252 (63.5)	
Female			145 (36.5)	
Number of children *n* (%)				
1			284 (71.5)	
2			112 (28.2)	
3			1 (0.3)	
Educational level *n* (%)				χ^2^ (4) = 15.89, *p* = 0.003
Elementary	4 (1)	3 (0.8)		
Junior high school	28 (7.1)	62 (15.7)		
High school	157 (39.6)	157 (39.6)		
Bachelor	204 (51.5)	171 (43.2)		
Master	3 (0.8)	3 (0.8)		
Occupation *n* (%)				χ^2^ (1) = 45.80, *p* < 0.001
Unemployed or housekeeper	1 (0.3)	46 (11.6)		
Employed	396 (99.7)	351(88.4)		
Living area *n* (%)				
Urban			271 (68.3)	
Rural			126 (31.7)	
Monthly income a (CNY) *n* (%)				χ^2^ (3) = 119.57, *p* < 0.001
0–3000	7 (1.8)	71 (18)		
3001–6000	142 (35.8)	205 (52)		
6001–10,000	202 (50.9)	112 (28.4)		
>10,000	45 (11.3)	6 (1.5)		
Family monthly income b (CNY) *n* (%)				χ^2^ (3) = 0.25, *p* = 0.969
0–5000			3 (0.8)	
5001–10,000			158 (39.8)	
10,001–15,000			105 (26.4)	
>15,000			131 (33.0)	
Cost of caretaking of child(ren) with ASD (Year- CNY) c *n* (%)				χ^2^ (3) = 0.04, *p* = 0.998
None			15 (3.8)	
1–30,000			324 (81.6)	
30,001–60,000			49 (12.3)	
>60,000			9 (2.3)	
Time spent for caretaking of children each day (hours) *n* (%)				χ^2^ (3) = 176.66, *p* < 0.001
0–3	176 (44.3)	40 (10.1)		
6–8	174 (43.8)	165 (41.7)		
9–10	38 (9.6)	119 (30.1)		
>10	9 (2.3)	72 (18.2)		
Duration of current marriage (Years) *n* (%)				χ^2^ (3) = 0.27, *p* = 0.965
3–5	3 (0.8)	4 (1)		
6–8	4 (1)	5 (1.3)		
9–10	39 (9.8)	39 (9.8)		
>10	351 (88.4)	349 (87.9)		
Have been married before *n* (%)				χ^2^ (1) = 0.82, *p* = 0.365
No	384 (97)	388 (98)		
Yes	12 (3)	8 (2)		
Type of marriage *n* (%)				
Self-deliberated			228 (57.9)	
Arranged			166 (42.1)	
Interpersonal Style *n* (%)				χ^2^ (7) = 7.82, *p* = 0.348
Domineering/Controlling	44 (11.1)	42 (10.6)		
Vindictive/Self-centered	67 (16.9)	77 (19.4)		
Cold/Distant	83 (20.9)	84 (21.2)		
Socially inhibited	44 (11.1)	45 (11.3)		
Nonassertive	23 (5.8)	33 (8.3)		
Overly Accommodating	24 (6.0)	16 (4.0)		
Self-Sacrificing	59 (14.9)	41 (10.3)		
Intrusive/Needy	52 (13.1)	59 (14.9)		
Interpersonal complementarity *n* (%)				
Carson			273 (68.8)	
Similarity			321 (80.9)	
Wiggins			270 (68.0)	
Psychological measurement, Mean (SD)				
Autism Behavior Checklist	54.70 (4.83)	54.52 (5.47)		t = 0.49, *p* = 0.623
Perceived family support score	18.02 (3.77)	18.19 (3.52)		t = 0.66, *p* = 0.512
Couple Satisfaction score	65.27 (10.82)	65.60 (10.87)		t = 0.43, *p* = 0.668

**Table 2 healthcare-10-01227-t002:** Bivariate correlations and descriptive statistics.

Variables	Actor/Partner	Husband’s Relationship Satisfaction	Wife’s Relationship Satisfaction
Age (Years), 23–45	Actor	−0.083	0.001
	Partner	0.046	−0.084
Educational level	Actor	0.033	0.003
	Partner	0.098	−0.014
Monthly income (CNY)	Actor	0.027	0.002
	Partner	0.114 *	−0.004
Time spent	Actor	−0.100 *	−0.047
	Partner	−0.077	−0.045
Autism Behavior Checklist	Actor	−0.24 **	0.085
	Partner	0.052	−0.207 **
Perceived Family support	Actor	0.385 **	0.033
	Partner	0.125 *	0.451 **
Age of first child (Years), 7–14	Dyad	−0.024	0.045
Family monthly income (CNY)	Dyad	0.020	0.017
Cost of caretaking (Year- CNY)	Dyad	0.098	0.045
Duration of marriage (Years)	Dyad	−0.067	0.055
Type of Marriage	Dyad	0.119 *	0.037
Residence area	Dyad	−0.030	−0.107 *
Interpersonal complementarity			
Carson	Dyad	0.001	0.099 *
Similarity	Dyad	0.018	0.149 **
Wiggins	Dyad	−0.048	0.065

* *p* < 0.05, ** *p* < 0.01.

**Table 3 healthcare-10-01227-t003:** Separate effect estimates for the actor–partner interdependence.

Variable	Role	Effect	Estimate	Lower	CI	*p*-Value	Beta (0)	Beta(s)	r
Couple satisfaction	Husbands	Intercept	82.849	65.685	<0.001	<0.001			
	Wives		60.791	43.870	77.713	<0.001			
Monthly Income	Husbands	Actor	−2.285	−3.762	−0.808	0.003	−0.159	−0.153	−0.154
		Partner	−0.121	−1.507	1.266	0.865	−0.008	−0.009	−0.009
		K	0.053	−0.655	0.965				
	Wives	Actor	−2.752	−4.200	−1.304	<0.001	−0.192	−0.199	−0.186
		Partner	0.129	−1.289	1.547	0.858	0.009	0.009	0.009
		K	−0.047	−0.622	0.582				
Time spent	Husbands	Actor	−2.215	−3.399	−1.031	<0.001	−0.189	−0.181	−0.185
		Partner	−1.128	−2.236	−0.020	0.046	−0.096	−0.101	−0.102
		K	0.509	0.007	1.509				
	Wives	Actor	−1.852	−2.949	−0.754	<0.001	−0.158	−0.165	−0.167
		Partner	−0.681	−1.855	0.493	0.256	−0.058	−0.055	−0.058
		K	0.368	−0.267	1.556				
Severity of autistic behaviors	Husbands	Actor	−0.635	−0.833	−0.437	<0.001	−0.302	−0.278	−0.307
		Partner	0.073	−0.099	0.245	0.404	0.035	0.037	0.044
		K	−0.115	−0.414	0.160				
	Wives	Actor	−0.334	−0.507	−0.161	<0.001	−0.159	−0.168	−0.191
		Partner	0.114	−0.074	0.302	0.235	0.054	0.051	0.060
		K	−0.341	−1.139	0.240				
Family Support	Husbands	Actor	1.138	0.886	1.390	<0.001	0.383	0.398	0.413
		Partner	0.337	0.065	0.609	0.015	0.113	0.110	0.125
		K	0.296	0.055	0.571				
	Wives	Actor	1.471	1.200	1.743	<0.001	0.495	0.475	0.476
		Partner	−0.008	−0.261	0.244	0.948	−0.003	−0.003	−0.000
		K	−0.006	−0.177	0.169				
Living	Husbands		−1.755	−3.805	0.294	0.094	−0.075	−0.076	−0.090
	Wives		−2.576	−4.647	−0.504	0.015	−0.111	−0.110	−0.120
Type of Marriage	Husbands		2.047	0.098	3.995	0.040	0.093	0.093	0.102
	Wives		1.487	−0.436	3.410	0.130	0.068	0.068	0.078
Interpersonal Complementarity (similarity)	Husbands		−1.346	−3.687	0.995	0.260	−0.049	−0.049	−0.064
	Wives		2.327	−0.042	4.697	0.055	0.084	−0.084	0.100

**Table 4 healthcare-10-01227-t004:** Overall effect estimates for the actor–partner interdependence model.

Variables	Effect	Estimate	Lower	Upper	*p*-Value	Beta	r
Couple satisfaction	Intercept	71.820	59.199	84.441	<0.001		
Income	Actor	−2.518	−3.548	−1.489	<0.001	−0.176	−0.170
	Partner	0.004	−0.982	0.990	0.993	0.000	−0.000
	K	−0.002	−0.432	0.424			
Time	Actor	−2.033	−2.835	−1.232	<0.001	−0.174	−0.176
	Partner	−0.905	−1.706	−0.103	0.027	−0.077	−0.079
	K	0.445	0.052	1.014			
Severity of behaviors	Actor	−0.485	−0.616	−0.353	<0.001	−0.229	−0.253
	Partner	0.094	−0.033	0.221	0.149	0.045	0.052
	K	−0.193	−0.497	0.070			
Family Support	Actor	1.305	1.120	1.489	<0.001	0.439	0.446
	Partner	0.164	−0.020	0.349	0.082	0.055	0.065
	K	0.126	−0.015	0.271			
Living area		−2.165	−3.628	−0.703	0.004	−0.093	−0.105
Type of Marriage		1.767	0.388	3.146	0.012	0.081	0.090
Interpersonal complementarity (similarity)		0.491	−1.167	2.148	0.562	0.018	0.018

## Data Availability

The datasets used and/or analyzed during the current study are available from the corresponding author upon reasonable request.

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
