# Peer review of "Marital Satisfaction and Perceived Family Support in Families of Children with Autistic Spectrum Disorder: Dyadic Analysis"

_healthcare, 2022, doi:10.3390/healthcare10071227_

Round 1

Reviewer 1 Report

Being a researcher in the autism field, I enjoyed reading your manuscript. The topic you covered is an important one since most research focuses on the autistic child rather than the parents.

Overall, your article is well written and well presented. I have recommendations for minor edits. For example, line 65 should read "Parents of children with ASD..." And line 379,  the word "reduces" should be "reduced" since your sentence is in past tense. And line 386, "Severity" should not be capitalized.

Line 81, please revise "problems" and "disturbed behaviors" to challenging behaviors. do a read through for spelling and grammar check.

Under Procedure, line 141 please delete "anonymity". There is a difference between confidential and anonymous. Your study is confidential since you know the parents completing the surveys (since you ask for medical records which I am assuming have names listed). Anonymous, on the opposite, means you do not know the participants since they did not provide any personal identifying information. Hence, line 141 should read, "Questionnaire data were kept confidential to protect the identity of the participants."

Under Conclusion, line 442-443 does not make sense. I would revise the wording to read, "...were associated with degree of relationship satisfaction."

To conclude, I would like to make an advocacy plea. Rather than saying "autism symptoms", revise to "severity of behaviors." Also, your title has the term "autistic children", I recommend staying with the same term throughout the entire paper rather that switching between "children with ASD" and "autistic".

Author Response

Department of Psychiatry, Faculty of Medicine, Chiang Mai University, Chiang Mai,

Kingdom of Thailand. 50200

23 June 2022

Dear Editor,

Re: Marital Satisfaction and Perceived Family Support in Families of Children with Autistic Spectrum Disorder: Dyadic Analysis, dated 8th May 2022

            I would like to thank the editor and the reviewers for their useful comments and suggestions. Please see below our point-by-point responses to the reviewers’ comments.

Reviewer Comments:
Reviewer #1

Being a researcher in the autism field, I enjoyed reading your manuscript. The topic you covered is an important one since most research focuses on the autistic child rather than the parents.

Authors’ Response: Thank you for your encouragement.

Comment #1 - Overall, your article is well written and well presented. I have recommendations for minor edits. For example, line 65 should read "Parents of children with ASD..." And line 379, the word "reduces" should be "reduced" since your sentence is in past tense. And line 386, "Severity" should not be capitalized.

Authors’ Response: Thank you. We have revised it as suggested. They are now appeared on line 61, 381, 390, and highlighted in blue color.

Comment #2 -Line 81, please revise "problems" and "disturbed behaviors" to challenging behaviors. do a read through for spelling and grammar check.

Authors’ Response: Thank you. We have revised the words to ‘challenging behaviors’ and did the spelling and grammar check. Please see line 77.

Comment #3- Under Procedure, line 141 please delete "anonymity". There is a difference between confidential and anonymous. Your study is confidential since you know the parents completing the surveys (since you ask for medical records which I am assuming have names listed). Anonymous, on the opposite, means you do not know the participants since they did not provide any personal identifying information. Hence, line 141 should read, "Questionnaire data were kept confidential to protect the identity of the participants."

Authors’ Response: Thank you. We agree that ‘identity’ is more suitable, and we have revised it. Please see line 139.

Comment #4 -Under Conclusion, line 442-443 does not make sense. I would revise the wording to read, "...were associated with degree of relationship satisfaction."

Authors’ Response: Thank you. We have revised as suggested. Please see line 444.

Comment #5 -To conclude, I would like to make an advocacy plea. Rather than saying "autism symptoms", revise to "severity of behaviors." Also, your title has the term "autistic children", I recommend staying with the same term throughout the entire paper rather that switching between "children with ASD" and "autistic".

Author Response: Thank you for your suggestion. We have revised the word ‘autism symptoms’ to ‘severity of autistic behaviors’ in the whole manuscript. And we also use the same term ‘children with ASD’ in the whole manuscript as suggested.

Reviewer #2

Comment #1 - Throughout, English needs improvement; I only have time to note some of the needed corrections. Line 33, suggest "implications for interventions"

Authors’ Response: Thank you. We have revised the words to ‘implications for interventions’ Please see line 30 (highlighted in red color).

Comment #2 - Line 44, suggest "and the marital relationship"

Authors’ Response: Thank you. We have revised the words to ‘and the marital relationship’. Please see line 41.

Comment #3- Line 60, suggest "while a good marital"

Authors’ Response: Thank you. We have revised the words to ‘while a good marital’. Please see line 56.

Comment #4- Line 61, suggest "in the challenges"

Authors’ Response: Thank you. We have revised the words. Please see line 57.

Comment #5- Line 63, "support has played an important and crucial role as a resource in the"

Authors’ Response: Thank you. We have revised the words. Please see line 60.

Comment #6- Line 124  How did COVID impact the methods used?

Authors’ Response: Thank you for this question. During COVID-19 and a strict policy of China, online research is the only possible way to conduct research. We have added the related information, which now reads ‘The research employed a cross-sectional survey, conducted online between December 2021 and January 2022 due to COVID-19 and the lockdown policy in mainland China.’ Please see line 120 and 121.

Comment #7-Line 140.  Does this mean each couple were provided with two surveys and each spouse filled out their own independently?  As it stands, the meaning is unclear.

Authors’ Response: Yes, that’s correct. We have revised the words to explain more clearly as  ‘Each couple were provided with two surveys and each spouse filled out their own independently.’ Please see line 137 and 138.

Comment #8-Line 181   With what other marital satisfaction scales has the CSI been validated in China or elsewhere?

Authors’ Response: Thank you for pointing this out. We have not found that the Chinese version of CSI-16 has been validated. That’s why we checked its internal consistency using Cronbach’s alpha in the pilot Chinese sample for conducting the research.

Comment #9- Line 183  "The sample questions included, "Our..."

Authors’ Response: Thank you. We have revised the words. Please see line 185.

 Comment #10-Line 198   "from 0.70 to 0.84."

Authors’ Response: Thank you. We have revised as suggested. Please see line 202.

Comment #11- Line 211   Please provide a few references for modern versions of exchange theory

Authors’ Response: Thank you. We have added the references as suggested as shown below

  1. Christensen, A.; Atkins, D.C.; Berns, S.; Wheeler, J.; Baucom, D.H.; Simpson, L.E. Traditional versus integrative behavioral couple therapy for significantly and chronically distressed married couples. J Consult Clin Psychol 2004, 72, 176-191, doi:10.1037/0022-006x.72.2.176.
  2. Molm, L.D. The Structure of Reciprocity. Social Psychology Quarterly 2010, 73, 119-131, doi:10.1177/0190272510369079.
  3. Lawler, E.J.; Thye, S.R.; Yoon, J. Social Exchange and Micro Social Order. American Sociological Review 2008, 73, 519-542, doi:10.1177/000312240807300401.

Comment #12- Line 258   "made their decision to choose their own partner"

Authors’ Response: Thank you.  We have revised as suggested. Please see line 263.

Comment #13-Line 259  I think you should cite the percentage here (88.4/87.9)

Authors’ Response: Thank you for your suggestion. We have added the related percentage. Please see line 264-265.

Comment #14- Table 1, number of children.  You have 1, 2, and > 3; why not 1, 2, 3, and > 3?

Authors’ Response: We apologize for the mistake.  We have revised it to 1, 2, and 3. Please see table1.

Comment #15- Line 265.  Please clarify the relationship between CNY and RMB.

Authors’ Response: Thank you for pointing this out. CNY and RMB is the same meaning in China. RMB is the traditional term for Chinese currency. We have changed ‘RMB’ to ‘CNY’ in the whole manuscript.

Comment #16- Line 284 “There were four outliers in the dataset which were removed"

Authors’ Response: Thank you. We have revised the text. Please see line 288.

Comment #17- Line 314   This line is confusing - were they different but not significant or not different and not significant?

Authors’ Response: We apologize for the confusing sentence. We have revised the words to express more clearly ‘). Two actor effects were not significantly different, Z = 1.76 (p = .079).’ Please see line 315 and 316.

Comment #18 Line 346  Is p correct for z = 1.49?

Authors’ Response: Thank you. We have revised the p value. Please see line 348.

Comment #19 Line 348 seems incomplete, which tests, for what?

Authors’ Response: We have revised as follow, “Two effects were significant different, Z = 2.15 (p = .032).” Please see line 350

Comment #20 Line 366   delete "about"

Authors’ Response: Thank you. We have deleted the word ‘about’. Please see line 368.

Comment #21 Line 386   "severity"

Authors’ Response: Thank you. We have revised as suggested. Please see line 390.

Comment #22 Line 437   suggest "western cultures".

Authors’ Response: Thank you for your suggestion. We have revised the word. Please see line 438.

Comment #23 References; some seem lacking completeness (e.g., 1, 12, 17, 26, 40, 41, 43, 47,  there seem to be several where only the first page is provided rather than the range of pages.1. The 5th and 6th paragraph of the introduction share a same thesis. It'd be better to combine them into one paragraph.
Author Response: Thank you very much for pointing this out. It’s very helpful. We have combined the 5th and 6th paragraphs and rewritten all references. Please see reference’s part.

Hopefully, our revisions are sufficient and satisfy the editor and reviewers. We have also corrected many other errors and misspellings throughout the manuscript in green and purple texts. Thank you for your consideration and positive attitude in publishing our study. We look forward to hearing from you soon.

Best regards,

Tinakon Wongpakaran

Bijing He

Nahathai Wongpakaran

Danny Wedding

Reviewer 2 Report

1.  Throughout, English needs improvement; I only have time to note some of the needed corrections.

2.  Line 33, suggest "implications for interventions"

3.  Line 44, suggest "and the marital relationship"

4.  Line 60, suggest "while a good marital"

5.  Line 61, suggest "in the challenges"

6.  Line 63, "support has played an important and crucial role as a resource in the"

7.  Line 124  How did COVID impact the methods used?

8.  Line 140.  Does this mean each couple were provided with two surveys and each spouse filled out their own independently?  As it stands, the meaning is unclear.

9.  Line 181   With what other marital satisfaction scales has the CSI been validated in China or elsewhere?

10.  Line 183  "The sample questions included, "Our..."

11.  Line 198   "from 0.70 to 0.84."

12.  Line 211   Please provide a few references for modern versions of exchange theory

13.  Line 258   "made their decision to choose their own partner"

14.  Line 259  I think you should cite the percentage here (88.4/87.9)

15.  Table 1, number of children.  You have 1, 2, and > 3; why not 1, 2, 3, and > 3?

16.  Line 265.  Please clarify the relationship between CNY and RMB.

17.  Line 284  "There were four outliers in the dataset which were removed"

18.  Line 314   This line is confusing - were they different but not significant or not different and not significant?

19.  Line 346  Is p correct for z = 1.49?

20.  Line 348 seems incomplete, which tests, for what?

21.  Line 366   delete "about"

22.  Line 386   "severity"

23.  Line 437   suggest "western cultures".

24.  References; some seem lacking completeness (e.g., 1, 12, 17, 26, 40, 41, 43, 47,  there seem to be several where only the first page is provided rather than the range of pages.

Author Response

(The authors gave the same response as above.)

Reviewer 3 Report

I read the article carefully. I think the purpose of the research is very interesting. In my opinion, it has been implemented to a very good extent. As a value of research, I consider a very large research group. Congratulations!

Author Response

Thank you very much.
